# Brain connectivity fingerprinting and behavioural prediction rest on distinct functional systems of the human connectome

Maron Mantwill [1,2,5✉], Martin Gell[2,3,4,5], Stephan Krohn[1,2] & Carsten Finke [1,2]

The prediction of inter-individual behavioural differences from neuroimaging data is a rapidly evolving field of research focusing on individualised methods to describe human brain organisation on the single-subject level. One method that harnesses such individual signatures is functional connectome fingerprinting, which can reliably identify individuals from large study populations. However, the precise relationship between functional signatures underlying fingerprinting and behavioural prediction remains unclear. Expanding on previous reports, here we systematically investigate the link between discrimination and prediction on different levels of brain network organisation (individual connections, network interactions, topographical organisation, and connection variability). Our analysis revealed a substantial divergence between discriminatory and predictive connectivity signatures on all levels of network organisation. Across different brain parcellations, thresholds, and prediction algorithms, we find discriminatory connections in higher-order multimodal association cortices, while neural correlates of behaviour display more variable distributions. Furthermore, we find the standard deviation of connections between participants to be significantly higher in fingerprinting than in prediction, making inter-individual connection variability a possible separating marker. These results demonstrate that participant identification and behavioural prediction involve highly distinct functional systems of the human connectome. The present study thus calls into question the direct functional relevance of connectome fingerprints.

[1] Charité-Universitätsmedizin Berlin, Department of Neurology, Berlin, Germany. [2] Humboldt-Universität zu Berlin, Faculty of Philosophy, Berlin School of Mind and Brain, Berlin, Germany. [3] Institute of Neuroscience and Medicine (INM-7: Brain & Behaviour), Research Centre Jülich, Jülich, Germany. [4] Department of Psychiatry, Psychotherapy and Psychosomatics, Medical Faculty, RWTH Aachen University, Aachen, Germany. [5] These authors contributed equally: Maron Mantwill, Martin Gell. ✉email: maron.mantwill@charite.de

The mapping of individual cognitive and behavioural performance to neurological patterns, and the identification of robust disease biomarkers are primary goals of neuroscience[1,2]. Indeed, the ability to predict individual cognitive performance or identify neurological disease markers is often regarded as an important development towards a more individualised behavioural neuroscience[3,4]. However, this focus on the individual requires a shift from group-level to single-subject analyses, moving the focus from finding average differences between groups into a more mechanistic understanding of the underlying processes[5], accommodating idiosyncratic differences between individuals.

Connectome fingerprinting represents one such individualised and powerful approach to single-subject analysis[6,7]. In connectome fingerprinting, individual participants can be reliably identified within large data sets with accuracies exceeding 90%, based on the discriminatory power of individual functional connectomes. Although the method can be used as a measure of the uniqueness and reliability of individual functional connectomes[8–10], its large appeal to the community is likely more practical: the possibility of a relation between connectome fingerprinting and behaviour, individual traits, or clinical markers[5]. Since its conception, connectome fingerprinting has raised the intriguing question whether distinctive individual connectivity signatures are also functionally relevant to variation in behaviour[7]. Subsequent literature on fingerprinting is ripe with parallel investigations of individual identifiability and prediction of behaviour, using static functional connectivity[8,11,12], dynamical functional connectivity[13], structural connectivity[14] and structural features like cortical thickness or myelin[12] as the basis for identification and prediction. Indeed, those resting-state networks that show the highest inter-individual variability—such as the frontoparietal, default mode and dorsal attention network[15–17]—have also been shown to contain highly discriminatory features in fingerprinting[6,7,18–20] and are at the same time commonly associated with behaviorally and clinically relevant variability[21–23]. In consequence, discriminatory fingerprinting signatures and inter-individual variability in behaviour have consistently been interpreted or assumed to be related[7,12–14,24–26].

However, such interpretations of the functional relevance of connectome fingerprinting commonly rely on visual inspection of the network-level organisation (i.e., the relative distributions of predictive or discriminatory connections in different resting-state networks), whereas both connectome fingerprinting and behavioural prediction ultimately rest on individual edges. As such, claims about a link between behaviour and fingerprinting draw upon aggregated data instead of the underlying region-to-region connections. This is further paralleled by a lacking analytical explanation of the assumed relationship. Therefore, although previous findings have revealed suggestive network-level patterns pointing to a potential overlap between discriminatory and predictive features of the functional connectome, a detailed, multi-layered statistical analysis is necessary to investigate if such a relationship can be verified empirically.

Here, we investigate the purported relationship between predictive and discriminatory resources of connectome fingerprints in detail. We first replicate the analysis from Finn et al.[7], confirming a suggestive overlap based on visual inspection. However, a systematic examination of this overlap in diverse behaviours shows that discriminatory connectome features and connections predictive of behaviour are unrelated on different levels of organisation: single edges, network level and topographical distribution. These findings are robust with respect to different parcellation schemes and prediction algorithms. Together, our results suggest an alternative perspective on the relationship between fingerprinting and behavioural prediction, resting on edge-level variability.

## Results

We first replicated the high fingerprinting accuracies and within-network overlap between predictive and discriminatory features of the connectome observed previously[7]. Next, we investigated the overlap between the features of interest in fingerprinting and prediction on edge-by-edge, network-by-network, and a large-scale topographical level. Here, we only present the results for fluid intelligence, language comprehension, and grip strength for the positive models of the CPM[27] framework (see Methods for details), with a feature selection threshold of $p < 0.01$. However, results equivalently hold for negative models as well as for other feature selection thresholds ($p < 0.001$, $0.005$, $0.05$), and can be found in the supplements (Suppl. Figs. 1–3).

**Connectome fingerprinting and network distribution of within-network connections**. We observed high fingerprinting accuracy of 96.8% (328/339, permutation-derived $p < 0.001$ against chance) when identifying individuals from session 1, and 97.3% (330/339, $p < 0.001$) when identifying individuals from session 2. Focusing on within-network connections, we observed strong involvement of highly discriminatory edges from the medial frontal (MFN), frontoparietal (FPN), and default mode network (DMN) as well as minor involvement of the subcortical-cerebellar network (SCN) (Fig. 1a). These results closely resemble previous findings with comparable accuracies and within-network contributions[7].

**Within-network connections and behavioural prediction**. We found a significant correlation between the measured values and the predicted values of fluid intelligence ($r(316) = 0.22$, $p < 0.001$; Fig. 1b). The within-network connections that were most often selected as features were found in the MFN, FPN and SCN (Fig. 1a). Akin to Finn et al.[7], networks supporting prediction resembled the networks displaying the highest proportion of discriminatory edges in fingerprinting, i.e., both the MFN and FPN contributed to fingerprinting and prediction of fluid intelligence (Fig. 1a). Further corroborating these findings, we found the DMN to be involved in fingerprinting but not in the positive prediction models, as was reported[7]. Motor Network (MN), Visual Network I (VN1), Visual Network II (VN2) and the Visual Association Network (VASN) did not strongly contribute to prediction or fingerprinting, as edges from those networks did not appear in the 99th percentile of discriminatory nor predictive edges.

Overall, we observed a significant correlation between measured and predicted values in 12 out of 30 psychometric variables (Suppl. Table 2 for all prediction results) including our two other variables of interest, language comprehension and grip strength (Table 1, all $p$ values permutation-derived, with $n = 1000$). When we examined the network contributions to the prediction of language comprehension, we found large involvement of within-network edges in the MFN, FPN and VASN, and more discrete involvement of the SCN within-network edges. Once more, these networks resembled those best discriminating between individuals. In strength prediction, VN2 and the VASN within-network edges were most predictive. In sum, within-network analyses of discriminatory and predictive edges seem to suggest a functional relevance of connectome fingerprints to inter-individual differences in higher-order cognitive functions such as fluid intelligence and language comprehension, but not grip strength, in line with the previous reports[7].

The above findings notwithstanding, if participant identification analysis is functionally relevant, some degree of overlap between discriminatory and predictive features would be expected beyond the mere resemblance of within-network contributions,

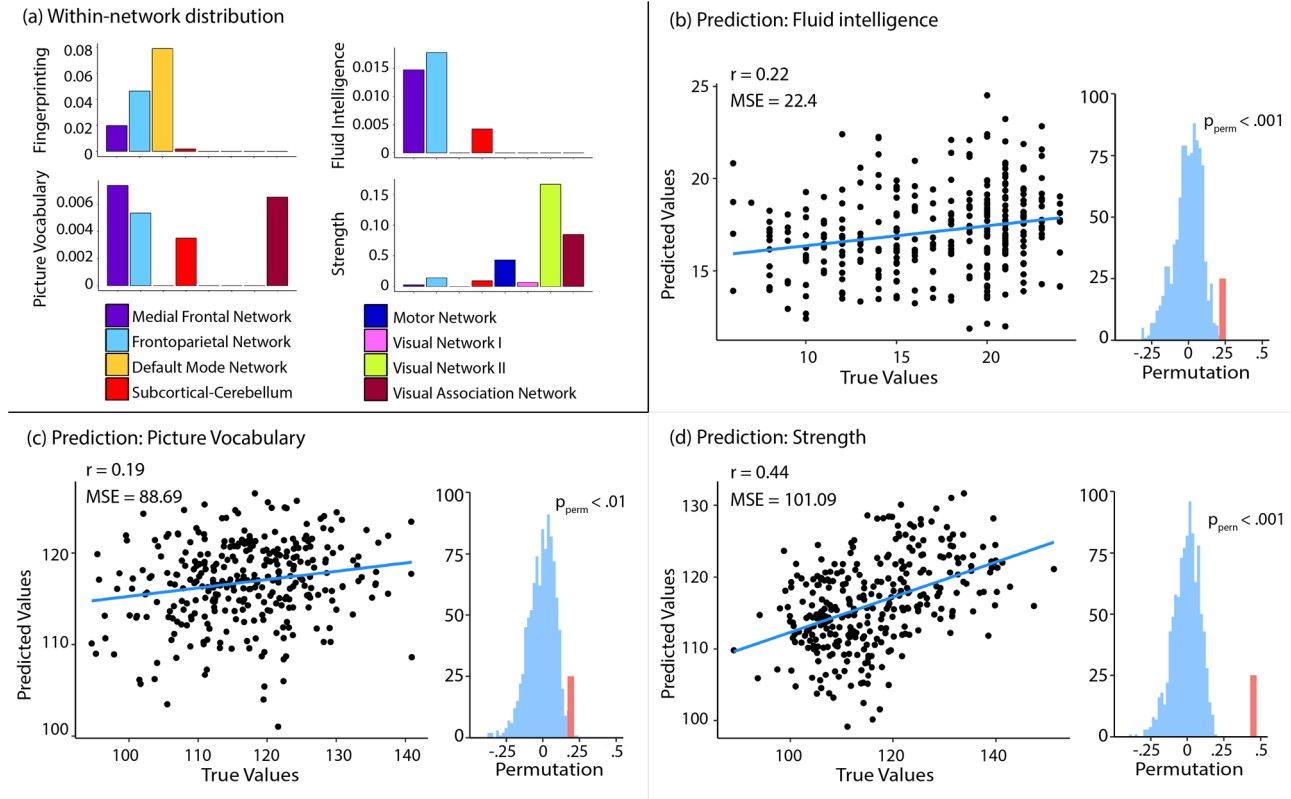

**Fig. 1 Within-network distribution of selected edges in fingerprinting and behavioural prediction. a** Visualises the percentage of selected edges for fingerprinting and for prediction within each network, adjusted for the total number of edges in each network. **b–d** shows the prediction results of three psychometric variables of interest. Language comprehension was evaluated using the picture vocabulary task.

e.g., overlap at the level of single edges, between-network connections, or the large-scale spatial distribution of discriminatory and predictive nodes.

**Single-edge overlap analysis**. We first investigated the overlap between highly discriminatory edges in fingerprinting and edges predictive of behaviour, without averaging or grouping these edges into resting-state networks (Table 2 and Fig. 2a). We found that the overlap between discriminatory and predictive edges did not exceed the chance level for any of the psychometric variables (Table 2).

Furthermore, if discriminatory edges are indeed relevant to the psychometric variables of interest—assuming that participant identification analysis was functionally relevant beyond the mere resemblance of within-network contributions—one should be able to predict these scores using the discriminatory edges directly. We tested this by modifying our prediction pipeline, replacing the feature selection step and instead directly applying the discriminatory edges in the training data of each CV fold. We then used this set of discriminatory edges to predict fluid intelligence, language comprehension, and grip strength in the test set. We found that predictions based on discriminatory edges could not significantly predict any of the three behavioural variables (Table 3), further corroborating that edges with high discriminatory potential are not related to behaviour on the single-edge level.

**Network overlap analysis**. Next, as individual functional connection weights have low reliability[28], we investigated the network distribution of edges, this time including both within- and between-network connections (Fig. 2b). For fingerprinting, we found a cluster of highly connected edges between as well as within the MFN, FPN and DMN, and to a lesser extent in the SCN. This was

in stark contrast to between-network connections found in psychometric prediction, which displayed a much more variable pattern. Most reliably, predictive features included connections between the DMN, the visual networks, and the rest of the brain, except for some within-network edges in the MFN and FPN for fluid intelligence. Furthermore, analysing the proportion of selected edges by network, we found that discriminatory edges did not significantly relate to fluid intelligence ($r(34) = -0.08$, $p = 0.620$), language comprehension ($r(34) = -0.15$, $p = 0.585$), nor grip strength ($r(34) = -0.39$, $p = 0.051$). Taken together, these findings suggest that even on a network level, highly discriminatory edges were not related to behaviour.

**Topological analysis of nodes with high degree of predictive edges**. Next, we investigated the overlap on a large-scale topological level. We found that discriminatory nodes (i.e., nodes with a high degree of discriminatory edges) clustered almost exclusively in the superior frontal, inferior parietal, and superior temporal regions (Fig. 3a). In line with our network-derived findings, predictive nodes displayed a more variable spatial distribution (Fig. 3b–d) and largely covered different parts of the cortex compared to discriminatory nodes. Corroborating these observations, there was no correlation between the spatial distribution of discriminatory nodes in fingerprinting and any behavioural prediction (Fig. 3b–d, right panel, all *p* values derived using spin permutation), again suggesting the spatial organisation of discriminatory nodes was not related to behaviour.

**Variability analysis**. Subsequently, we performed an exploratory analysis to investigate how edge properties relate to the divergence between discriminatory and predictive connectome features. Focusing on edge standard deviation of functional

**Table 1 Results for psychometric prediction.**

| Psychometric variable | Spearman correlation | p values | Participants (n) |
|---|---|---|---|
| Fluid cognition composite score | 0.22 | 0.002 | 318 |
| Crystalised cognition composite score | 0.21 | 0.001 | 320 |
| Total cognition composite score | 0.25 | <0.001 | 318 |
| Cognitive flexibility | 0.18 | 0.006 | 319 |
| Fluid Intelligence | 0.22 | <0.001 | 319 |
| Sustained attention (specificity) | 0.17 | 0.015 | 319 |
| Grip strength | 0.44 | <0.001 | 319 |
| Dexterity | 0.23 | <0.001 | 320 |
| Language/reading decoding | 0.19 | 0.003 | 320 |
| Language comprehension | 0.19 | 0.004 | 320 |
| Spatial orientation | 0.21 | 0.002 | 319 |
| Emotion recognition | 0.20 | 0.002 | 319 |

Last column contains the number of participants with complete data. All p values are permutation-derived and FDR-corrected for all behavioural predictions.

**Table 2 Overlap between highly discriminatory and predictive edges.**

| Psychometric variable | Number of predictive edges | Overlapping edges | Mean ± SD of permutation | p values |
|---|---|---|---|---|
| Cognitive flexibility | 212 | 0 | 1.46 ± 1.09 | 0.81 |
| Crystalised cognition composite score | 149 | 1 | 1.69 ± 1.31 | 0.5 |
| Fluid cognition composite score | 258 | 0 | 1.72 ± 1.07 | 0.88 |
| Total cognition composite score | 260 | 0 | 2.29 ± 1.27 | 0.92 |
| Dexterity | 359 | 0 | 0.74 ± 0.76 | 0.58 |
| Emotion recognition | 245 | 0 | 0.55 ± 0.72 | 0.46 |
| Fluid intelligence | 185 | 0 | 0.74 ± 0.81 | 0.55 |
| Language comprehension | 140 | 0 | 1.04 ± 1.03 | 0.65 |
| Language/reading decoding | 133 | 3 | 2.09 ± 1.27 | 0.13 |
| Sustained attention (specificity) | 133 | 1 | 0.87 ± 0.85 | 0.22 |
| Grip strength | 785 | 1 | 6.59 ± 2.24 | 0.99 |
| Spatial orientation | 179 | 2 | 2.46 ± 1.36 | 0.48 |

All p values were derived using degree-preserving permutations.

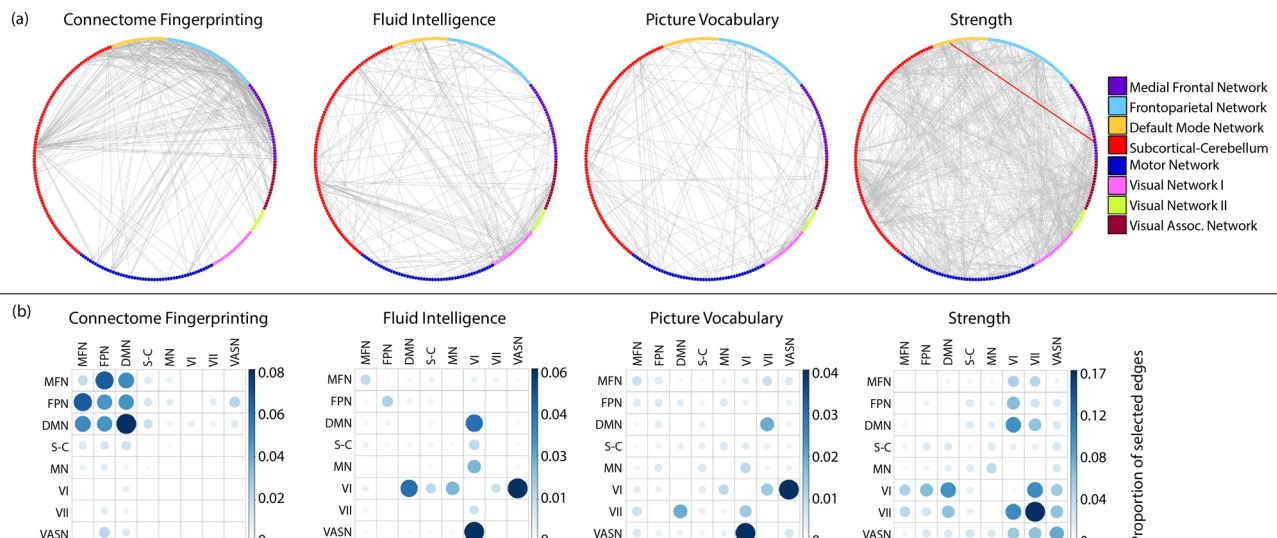

**Fig. 2 Single-edge and between-network overlap for fingerprinting and prediction. a** Grey lines mark highly discriminatory edges for fingerprints and predictive edges for behaviour, thresholded at the 99th percentile or *p* < 0.01 respectively. Red lines, only available for strength, visualise overlap. For fluid intelligence and language comprehension, no edges overlapped. **b** Shows the entire network-by-network matrix of the same selected edges, adjusted for a total number of edges.

connectomes, we discovered that discriminatory edges generally show high variability across participants (Fig. 4). Investigating this variability by comparing the 99th percentile of discriminatory edges in fingerprinting and the 99th percentile of edges with high standard deviation showed a strong and significant (122/358, *p* < 0.001) overlap, which increased even further when thresholding at the 98th percentile (179/358, *p* < 0.001) and the 95th percentile (286/358, *p* < 0.001; all *p* values permutation-derived,

Fig. 4a). Furthermore, edges used in prediction showed significantly lower variability for all three psychometric variables (Welch's $F(3, 409) = 675$, $p < 0.001$, est. $\omega^2 = 0.58$; post hoc comparisons using Games–Howell test all at $p < 0.001$, two-tailed, FDR-corrected) than discriminatory edges (Fig. 4b). Taken together, discriminatory edges in fingerprinting substantially overlap with edges showing higher variability in connectivity across participants, while edges predictive of behaviour are constrained to edges with intermediate variability.

**Validation analysis.** To make sure that our analysis generalised beyond a single prediction method, we repeated the analysis of edge-level overlap between discriminatory edges and predictive edges for all three behaviours, using Support Vector Regression instead of CPM. This independent prediction method corroborated the lack of overlap for all tested behaviours (Suppl. Table 3). In addition, we tested whether we could replicate our findings using different parcellation schemes. Focusing on the prediction of fluid intelligence, we observed significant correlations between predicted and measured intelligence scores using CPM with all three atlases (Brainnetome: $r(316) = 0.26$, $p < 0.001$, HCP: $r(316) = 0.18$, $p = 0.001$, AAL: $r(316) = 0.19$, $p < 0.001$). We also achieved fingerprinting accuracies of >90% for all atlases, with the HCP MMP 1 atlas resulting in accuracies of up to 99% (Suppl. Table 4). Our findings concerning a lack of overlap between discriminatory and predictive edges held true for between-network, anatomical and single-edge overlap (Brainnetome: $n = 3/301$, $p = 0.137$, HCP: $n = 3/646$, $p = 0.164$, AAL: $n = 0/67$,

$p = 0.269$) in all three parcellation schemes (Fig. 5a, b). We were also able to replicate the relationship between edge-variability and fingerprinting, showing a high overlap between the most discriminatory edges and edges with high standard deviation (Fig. 5c), as well as the significantly lower variability in edges predictive of behaviour (Fig. 5d).

## Discussion

In the present study, we show that fingerprinting signatures and behavioural prediction rest on highly distinct functional systems of the human connectome. We were able to replicate the seminal findings by Finn et al.[7], demonstrating high accuracy in participant identification with connectome fingerprinting as well as the importance of within-network edges in higher-order resting-state networks for both prediction and fingerprinting. These findings could be interpreted as supporting the functional relevance of fingerprinting signatures, that is, networks that best discriminate individuals from one another are also strongly involved in cognitive function[5,7]. However, these findings were restricted to a specific level of analysis (group-level within-network connections), motivating further exploration of the relationship.

We found evidence of a strong divergence between functional signatures supporting the prediction of behaviour and discriminability of the connectome. This held true on a network level when we considered both within as well as between-network connections, on the level of single edges, and on the level of the large-scale spatial organisation of discriminatory and predictive nodes. In addition, as a positive control, we directly used the edges with the highest discriminatory potential for the prediction of behaviour and found this to be unsuccessful, further corroborating our findings. To address the many degrees of freedom in the design of the analysis, we also show that our findings are highly robust against varying methodological choices. Specifically, we used four parcellation schemes, two prediction methods, and tested different feature selection thresholds. In sum, the results presented here suggest that discriminatory and predictive signatures of the human connectome rely on highly distinct functional systems.

**Table 3 Behavioural prediction results using discriminatory edges for model construction.**

| Psychometric variable | Correlation | *p* values | Participants (*n*) |
|---|---|---|---|
| Fluid intelligence | −0.19 | 0.069 | 319 |
| Grip strength | −0.04 | 1 | 319 |
| Picture vocabulary | 0.03 | 1 | 320 |

All *p* values are FDR-corrected.

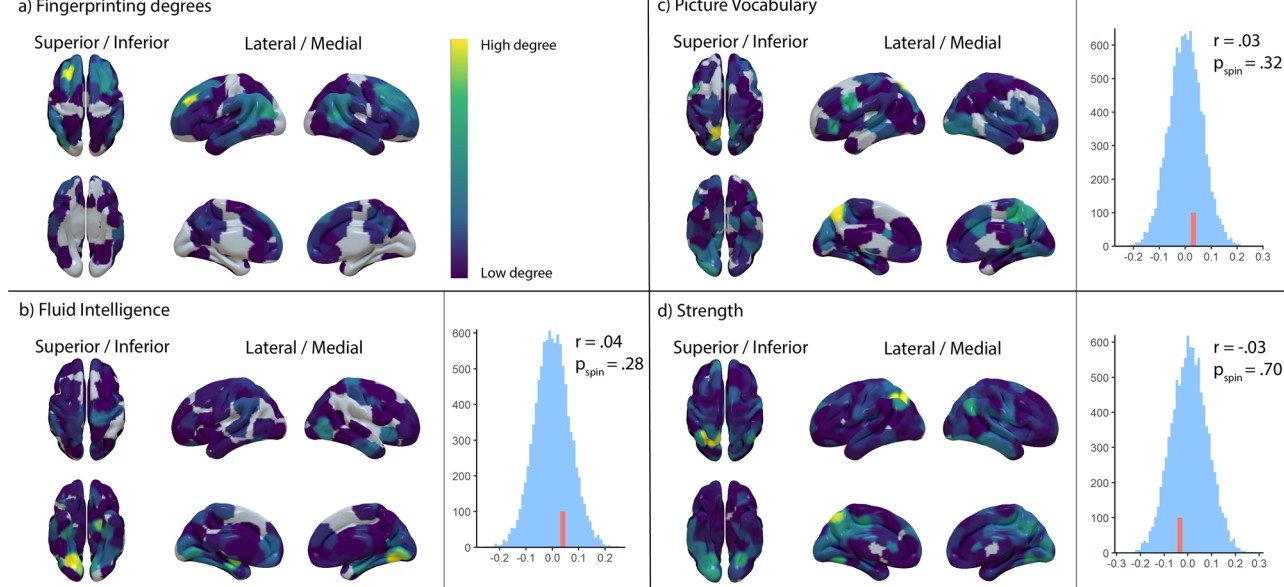

**Fig. 3 Spatial distribution of node degrees.** Distribution of node degrees for discriminatory nodes **a** and behaviourally predictive nodes (**b–d**) on the left. The edges underlying the node degrees are thresholded at the 99th percentile and $p < 0.01$ respectively. The right-hand side displays the spin permutation results, with red lines marking the empirical correlation of discriminatory and predictive nodes.

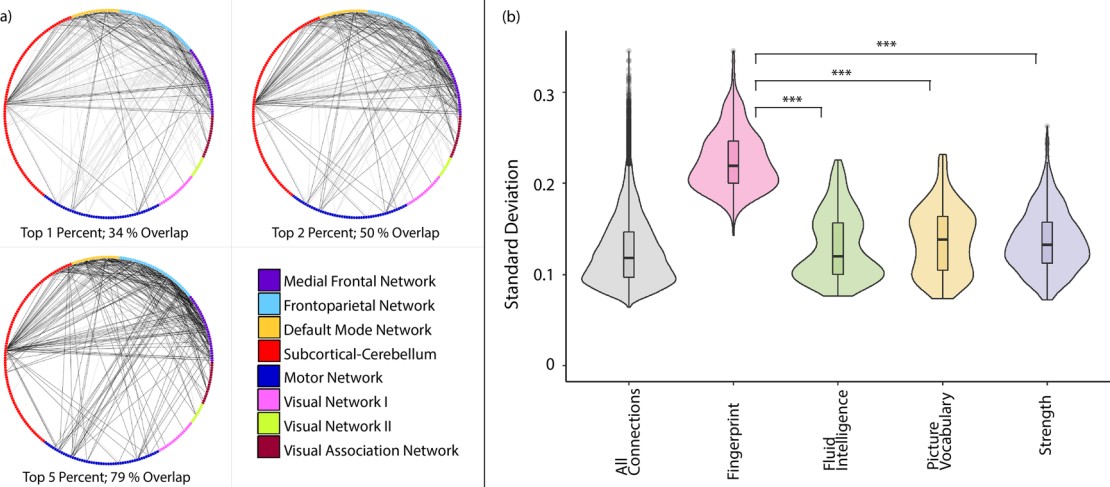

**Fig. 4 Overlap of discriminatory edges and high-variability edges. a** Black lines designate overlapping edges between the top one, two or five percent of discriminatory edges and high-variability edges. Grey lines depict non-overlapping discriminatory edges. **b** Shows the distribution of edge standard deviation across participants. In the boxplot, the middle line signifies the median, the lower and upper hinges correspond to the first and third quartiles and the upper and lower whisker represent 1.5 times the respective interquartile ranges. All points outside of whiskers are outliers. ***$p < 0.001$, FDR-corrected.

In this regard, our findings expand on the recent notion of a dichotomy between resources valuable for identification and behavioural prediction[12]. Although this dichotomy has been shown to apply with respect to different imaging modalities, here we show the separation to be present within a single modality. In this context, we show that variability of individual edges strongly distinguishes between fingerprinting and prediction signatures, and propose this variability to be at the root of the dichotomy. Discriminatory edges, but not predictive edges, showed a substantial overlap with edges that are highly variable across participants. The different mechanisms underlying fingerprinting and prediction might clarify these findings. In CPM, edges which show a significant correlation between functional connectivity and a behavioural outcome measure in the sample are selected as features. Thus, CPM is still a group-level procedure and requires edge variation to be linearly related to behaviour in order to be selected for prediction. In the framework of SVR, the feature selection is interdependent, i.e., the weight of a specific feature depends on the additional amount of information supplied beyond other features. Nevertheless, similarly to CPM, the edge variation also must be linearly related to variability in behaviour. In contrast, in fingerprinting, edges are selected based on intra-subject similarity, given sufficient inter-subject variability. Importantly, no group relationship is considered here. Prediction methods and fingerprinting thus relate differently to edge variability, i.e., edges selected in prediction need to covary with behaviour, whereas fingerprinting is impartial to the source of edge variability. Therefore, the high variability of functional connections selected in fingerprinting could result from a range of sources. For example, variations could stem from differences in functional network topology[17] or structural variabilities such as differences in cortical thickness[15] or folding patterns[29]. These differences might also result in stable variation in functional connectivity, whilst not necessarily relating to behaviour in a linear fashion. As such, increased variability in multimodal brain regions[15,30,31] may lead to a higher likelihood of individual-specific variation from different sources. Consequently, we observe clusters of discriminatory edges in these regions when averaging the discriminatory potential of individual edges over all participants. The significance of these functional variations is difficult to discern. Our results, however, point to the variation exploited during fingerprinting not being related to behaviour.

Further research will be necessary to establish whether edge variability also serves as a separating marker in other imaging modalities and whether the findings by Mansour and colleagues[12] might be supported by the relationship between signal variability, fingerprinting and behavioural prediction proposed here.

In the present work, we aimed to closely follow Finn et al.[7] in the preprocessing steps and used the same methods for the identification of participants, the prediction of behaviour and the extraction of high-value edges for fingerprinting. Furthermore, we mirrored the data analysis pipeline, initially focusing on within-network edges. Although we found functional connectivity to be a significant predictor of fluid intelligence with an accuracy similar to other published work[14,32,33], we did not achieve the high prediction scores reported by Finn et al.[7]. There are different possible explanations for this, one of them being our use of the unrelated sample from the HCP database. This sample has the advantage of being larger and thus more robust to overfitting, and it assures the independence between participants during cross-validation (CV)[34]. However, this independence might have influenced our prediction accuracies. Furthermore, we used 10-fold CV instead of leave-one-out CV[35].

Taken together, we show that participant identification and behavioural prediction from individual connectomes rely on highly distinct functional brain systems. This divergence raises the question of what the variability sustaining individual fingerprints ultimately relates to. Parsimony suggests that neurological variation should also be linked to phenotypic presentation, yet our results indicate that there is no simple one-to-one mapping between function and fingerprints. As such, further methodological development and conceptualisation will be necessary to deepen our understanding of individual functional signatures and their behavioural and biological significance.

## Methods
**Data set**. We used the unrelated subjects sample ($n = 339$, 156/183 m/f, ages 22–35) from the full release of the publicly available Human Connectome Project data set[36]. In our prediction analysis, we excluded participants that had missing behavioural data in a case-by-case fashion (Table 1). The HCP scanning protocol was approved by the local Institutional Review Board of Washington University in St. Louis, MO, USA, and informed consent was obtained from all participants, the details of which are described elsewhere[36]. In brief, for resting-state fMRI (rs-fMRI), whole-brain multiband gradient-echo-planar images were acquired on a 32-channel 3 T Siemens "Connectome Skyra" scanner with TR = 720 ms, TE = 33.1

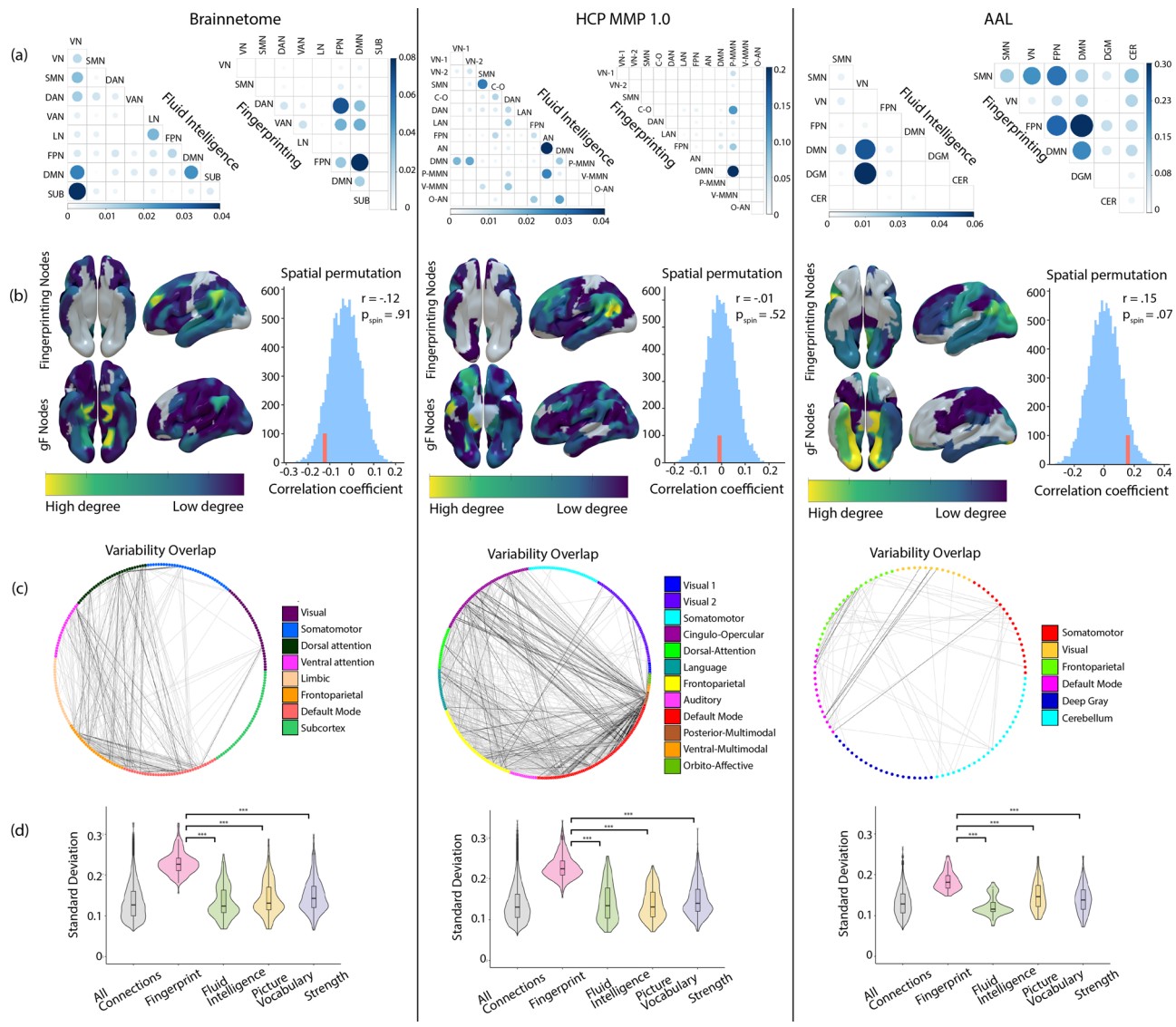

**Fig. 5 Control analyses for different functional atlases.** Results for Brainnetome, HCP MMP 1.0, and AAL atlases (left-to-right). From top to bottom, panels visualise **a** within and across-network connections, **b** spatial topology and spin permutation of nodes for fluid intelligence and fingerprinting, **c** the overlap of individual edges between highly discriminatory and high-variability edges, and **d** the distribution of edge standard deviations over participants in fingerprinting and behavioural prediction. In the boxplots, the middle line signifies the median, the lower and upper hinges correspond to the first and third quartiles and the upper and lower whisker represent 1.5 times the respective interquartile ranges. All points outside of whiskers are outliers. ***$p < 0.001$, FDR corrected.

ms, flip angle = 52 degrees, bandwidth = 2290 Hz/pixel, in-plane field of view = 208 × 180 mm², 72 slices, 2 mm isotropic voxels and 1200 volumes (14 min and 24 s). Rs-fMRI sessions were acquired left-to-right (LR) and right-to-left (RL). Furthermore, there were two separate rs-fMRI sessions for each individual ("rest 1" and "rest 2") acquired on two different days.

**rs-fMRI preprocessing.** We closely followed Finn et al.[7] in our preprocessing pipeline and used the minimally preprocessed rs-fMRI data set[37]. This included gradient distortion correction, motion correction, image distortion correction, registration to MNI standard space and intensity normalisation. We then used the CONN toolbox[38] for SPM12 to regress out 12 motion parameters (provided with the HCP data set under Movement_Regressors_dt.txt), mean time courses of white matter, CSF, and the global grey matter signal (approximating global signal). Linear trend was removed and the data were band-pass filtered (0.01–0.1 Hz). We did not perform any smoothing. The resulting voxel-wise time series were parcellated using four different atlases: for the main analyses, we used the Shen atlas with 268 nodes[7] (i.e., regions of interests (ROIs)); for validation, we used the Brainnetome atlas[39], the HCP multimodal parcellation (MMP) 1.0 atlas[40] and the AAL (Automated Anatomical Labelling) atlas[41]. For every parcellation scheme, we extracted the nodal time series by averaging over all voxels within the respective ROI.

The Shen atlas is a whole-brain atlas derived from functional connectivity and defined using a group-wise spectral clustering algorithm[42]. The three other atlases cover different levels of detail as well as different approaches to the definition of nodes. The AAL atlas provides an anatomy-based parcellation with 90 cortical and 26 cerebellar nodes[41]. The Brainnetome Atlas[39] is a whole-brain atlas containing a similar number of nodes to the Shen atlas (210 cortical and 36 subcortical nodes) and is defined using both anatomical and functional connections. HCP MMP 1.0 is a detailed cortical in-vivo parcellation with 360 nodes[40]. We acquired resting-state network definitions for all four atlases. The Shen and Brainnetome atlases provide resting-state network definitions for each node, and for the latter, these are based on the established Yeo-7 resting-state networks[43]. Since the Yeo-7 network definition does not assign subcortical nodes to a network, we created an eighth subcortical network. For the HCP MMP 1.0, we relied on network assignments by Ji et al.[44], partitioning the 360 nodes into 12 resting-state networks. The AAL nodes were split into five resting-state networks based on previously established network definition[45]. Here, we created a sixth cerebellar resting-state network including all cerebellar nodes.

**Functional connectome.** Individual functional connectomes were built as functional connectivity matrices calculated as the Pearson correlation between the time courses of all region-to-region pairs. In the framework of functional brain

networks[46], each ROI represents a network node, and the connection between two ROIs represents an edge in the network.

Each participant has two resting-state scans (LR and RL encoding) by session, thus creating four functional connectivity matrices by individual. We averaged the two matrices from one session (LR and RL), resulting in two final matrices (one per session) for every individual (matrix dimension: number of nodes by number of nodes, the exact number depending on the parcellation scheme). The final functional connectivity matrices were $z$ scored, and the upper triangle was vectorized.

**Functional connectome fingerprinting**. Fingerprinting was performed as in Finn et al.[7]. In brief, the functional connectome of a 'source' participant at timepoint t1 is used to identify the same participant at time point t2, referred to as 'target'. The target session is identified from a pool of functional connectomes containing both the target connectome as well as connectomes of other 'distractor' participants. Identification is performed by correlating the FC vector of the 'source' from one of the two scanning sessions (e.g., "rest 1") with the FC vectors of all 339 participants (including the 'target') in the other session (e.g., "rest 2"), resulting in 339 correlations. The participant with the highest correlation coefficient is picked and assigned a score of 1 if the picked participant matches the target identity (hit), and a score of 0 otherwise (miss). This procedure was applied for all possible session 1 to session 2 source-target pairs (i.e., 339 identifications) and then repeated once more for all session 2 to session 1 source-target pairs (again 339 identifications). At last, we performed a nonparametric permutation test with 1000 permutations to examine the statistical significance of our identification analysis. In each permutation, the target participant's and distractor session's identity were randomised and fingerprinting accuracy was recorded. $P$ values were then calculated as the proportion of randomly permuted instances exceeding the empirically observed accuracy over all permutations.

**Network and edge contributions to fingerprinting**. The analysis scripts with example data can be accessed in a public repository at (https://doi.org/10.5281/zenodo.4557011). To assess the contribution of different resting-state networks, we calculated the differential power of edges (i.e., node-to-node connections) using publicly available scripts by Finn and colleagues[7]. Differential power of an edge reflects an edge's 'uniqueness' and stability and thus its ability to differentiate an individual. First, we exclusively investigated the differential power of within-network edges in order to reproduce the original analysis. Here, we averaged the differential power of all within-network edges by their respective network, including edges with zero differential power. Second, we repeated the analysis including between-network connections. We then averaged the differential power between and within the different resting-state networks, creating a complete network-by-network matrix of differential power.

**Psychometric prediction**. For prediction, we used the Connectome-based Predictive Modelling approach (CPM)[27] and adapted the openly available script from: https://www.nitrc.org/frs/?group_id=51. Using this framework, we predicted 30 psychometric variables supplied in the HCP data set (see Suppl. Table 2). In our main analysis, we focus on three behavioural variables of interest and provide further results in the supplement. Specifically, we focused on the fluid intelligence score assessed by Penn Progressive Matrices used previously[7,47]. To broaden the scope of our analysis and examine psychometric variables unrelated to fluid intelligence, we selected two additional psychometric variables, grip strength and language comprehension (assessed using the Picture Vocabulary Task) based on their low correlation with fluid intelligence ($r = 0.02$ and $r = 0.20$, respectively; Suppl. Table 1 for all correlations).

Behavioural prediction with CPM consists of three steps: feature selection, model building and prediction. Features are selected by calculating the Pearson correlation between each edge in the training set and the psychometric variable. Edges are separated into correlated and anti-correlated edges and thresholded. Here, we tested: $p < 0.05$, 0.01, 0.005 and 0.001 following previous work[7]. Next, the thresholded correlated and anti-correlated edges are summed up, resulting in two summary values (a positive set and a negative set). Positive and negative summary values are used as a predictor of the measured cognitive variable in two linear regressions using least squares estimates. In the last step, positive and negative summary values are calculated for every participant in the test set using the same features identified during the feature selection step. The summary network strengths are then used to predict the cognitive variable. For a detailed description of CPM see[7,27].

We used 10-fold CV repeated 100 times, resulting in $10 \times 100$ measures of accuracy. To evaluate model accuracy, we collected the predicted cognitive scores for each participant in each repeat of our CV (i.e., 100 predictions per participant), and averaged across all repeats, following Nostro et al.[48]. To evaluate the significance of the relationship between the predicted and the measured scores, we performed permutation testing with 1000 permutations. In each permutation, we correlated the averaged predicted scores with the measured cognitive scores found in the HCP data. The $p$ value (right-tailed) was calculated by dividing the permutations that exceeded the non-permuted correlation value by the number of permutations plus one.

At last, we used an additional prediction method (support vector regression) to evaluate, whether the observed overlap of discriminatory and predictive edges held using different classifier types, independently of CPM. To this end, we repeated the above prediction procedure but removed the feature selection and model building steps used in CPM and instead used SVR for model building. SVR parameters were set at default values with no hyperparameter optimisation (linear kernel, $C = 0.75$, Regularisation = Lasso, Lambda = 0.0035). We extracted the highly predictive edges using the sorted SVR weights, which were thresholded and binarized for the subsequent overlap analysis. All edges selected in at least 80% of CV folds during prediction were used, resulting in the binarized matrix (described below).

**Overlap between differential power, predictive power, and high variability**. To perform our overlap analysis, we required binarized matrices of edges with high discriminatory potential and edges predictive of behaviour. For discriminatory potential, we thresholded the complete matrix of differential powers, setting all values below the 99th percentile to zero and all others to one, thus resulting in a sparse binary matrix. Highly predictive edges were thresholded by keeping all edges selected (i.e. significant at $p < 0.001$, 0.005, 0.01, or 0.05) in at least 80% of the CV folds during prediction, also resulting in a sparse binary matrix. In a final step, the overlap was obtained by overlaying the two resulting matrices and calculating the intersection of positive values. To assess the overlap of highly discriminatory and high-variability edges, we overlaid the differential power matrix with a matrix of standard deviations in functional connectivity across participants, thresholded at the 99th percentile. Other thresholds were also assessed (see Supplementary Figures 2 and 3 for visual representation and Supplementary Table 5 for overlap between the three tested psychometric variables and discriminatory edges).

Furthermore, we investigated the distribution of the standard deviations across all edges of the connectome, edges with high differential power, and edges predictive of the different psychometric variables. To assess if edge-to-edge overlaps were statistically larger than would be expected by chance, we performed a permutation test with 1000 permutations. In each permutation, we calculated the intersection between fingerprinting edges and a degree-preserving random matrix (preserving degrees of the predictive or SD matrix) using the Brain Connectivity Toolbox[46].

**Topographical localisation**. To localise regions important to either fingerprinting or the prediction of psychometric variables, we calculated the node degree for each region by summing up the number of connected edges in the sparse differential power matrix (in fingerprinting) and in the matrices of predictive edges (in psychometric prediction). To compare the topographical organisation found in fingerprinting and in prediction, we calculated the Spearman correlation between their node degrees and tested for significance of the topological overlap using spin permutation testing with 5000 permutations[49,50]. Spin permutation allows for correlational analyses of cerebral topology while conserving spatial data properties such as non-independence among neighbouring parcels.

**Statistics and reproducibility**. All fMRI resting-state data were preprocessed in Matlab2019b using the CONN toolbox (version 18b) for SPM12. Parcellation, functional connectivity ('corr' function), behavioural prediction and overlap analysis were also calculated in Matlab2019b using custom scripts (available here: https://doi.org/10.5281/zenodo.4557011). Fingerprinting analysis, statistics and visualisations were made using RStudio (R version 3.5.1).

Both minimally preprocessed brain-imaging data and psychometric variables were obtained from the Human Connectome Project website (https://db.humanconnectome.org/). All results reported here can be directly reproduced using the provided scripts, given additional preprocessing of the Human Connectome Project data (for details see above) or conceptually reproduced using functional connectivity matrices calculated from resting-state fMRI recordings available in other data sets. All intermediate output can be shared upon request.

**Reporting summary**. Further information on research design is available in the Nature Research Reporting Summary linked to this article.

## Data availability

Data are from the publicly available HCP repository and can be accessed at http://www.humanconnectomeproject.org/data/[36]. The list of unrelated participants used here can be accessed at https://wiki.humanconnectome.org/display/PublicData/S900+Unrelated+Subjects+CSV. Source data underlying the main figures are presented in Supplementary Data 1.

## Code availability

All scripts and resources utilised in the analysis reported here can be accessed in a public repository at (https://doi.org/10.5281/zenodo.4557011).

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

## Acknowledgements

Funded by the Deutsche Forschungsgemeinschaft (DFG, German Research Foundation), grant numbers FI 2309/1-1 (Heisenberg Programme) (CF), FI 2309/2-1 (CF, SK) and 269953372/GRK2150 (IRTG2150) (MG), by the Berlin School of Mind and Brain, Humboldt-Universität zu Berlin (MM) and by the Federal State of Berlin (MM).

## Author contributions

M.M. concept and design, data analysis, paper writing; M.G. concept and design, data analysis, paper writing; S.K. concept and design, paper writing; C.F. concept and design, supervision, paper writing.

## Funding

## Competing interests

The authors declare no competing interests.
