## [Transparent Peer Review File · Communications Biology]

Reviewers' comments:

Reviewer #1 (Remarks to the Author):

This a compelling article aiming to replicate the previous seminal work on fingerprinting (Finn et al. Nat Neuro 2015) and extend it to other tests (grip strength, language comprehension) that are not associated with fluid IQ used in the original report.

The authors used the unrelated HCP dataset (n=339) and four parcellation schemes (atlases: Shen/Brainnetome/HCP MMP 1.0/AAL), two prediction methods, and tested different feature selection thresholds.

The authors were able to replicate the original findings, albeit with a lower significance which they accurately address in the limitations. They were further able to show that connectivity patterns to identify individuals (discriminatory signatures) and to predict behaviour (predictive signatures) diverge and therefore conclude that there is no linear one-to-one association between behaviour and fingerprints.

The manuscript follows good practice in terms of open science and offers their codes online.

A minor comment would be that the healthy volunteers should not be referred to as 'subjects'.

Reviewer #2 (Remarks to the Author):

This manuscript examines the relationship between the functional connectivity features that are unique to an individual (ala functional fingerprinting) and those features that predict behavior.

The methods are generally robust, and the investigators have done a thorough job comparing across parcellation schemes, thresholds, and prediction algorithms, and performing cross-validation where possible.

The basic conclusion is that connectome fingerprints are not as functionally relevant as previously believed. But the "previously believed" part unfortunately represents a misreading of the literature.

To this reviewer's knowledge there is nothing in the literature that states that the connections that make an individual unique and identifiable (the functional fingerprinting part) are supposed to be the same connections linking functional organization to behavior (the prediction part).

Fingerprinting reveals that the connectivity matrices for individuals are both stable and unique to individuals (a finding that had been unexpected and not previously recognized). The Finn 2015 paper showed this. They also demonstrated that within this connectome there is information on the individual and connectome-based modeling supported this claim. These are 2 separate findings. Finn did show that the same networks fronto-parietal and default mode dominated in both fingerprinting and prediction but there was no claim that these were, or should, be the same features within those networks.

I don't believe that a close reading of the literature reveals any suggestions that the connections that yield brain fingerprints are the same as the connections that provide information on behavior.

Fingerprints have never been proposed as a method for personalized medicine.

Behavioral prediction supports the finding that connections vary across individuals in meaningful ways and may someday make individualized medicine possible. This says nothing about fingerprinting and should not be conflated with fingerprinting. The personalized medicine part comes in the sense that individuals can be characterized through predictive models. This work does not dispute that.

The authors have the correct and relevant explanation for why fingerprinting and predictive modeling

are different as they explain in the discrimination/variability discussion in lines 428-437. These are different methods not designed to reveal the same features – and they don't.

The thorough investigation here of the edges involved in the models and in identification is potentially interesting as is the edge-level variability work, and the node topography findings. But the current framing of the manuscript represents a misreading of the literature.

This reviewer disagrees with the assumptions behind the following statements:

Line 33 – No one has suggested that connectome fingerprints are the functionally relevant features.

Line 34, 35 – The results in this manuscript support that individual behavioral prediction is supported by connectome based predictive modeling and thus no re-evaluation is necessary. The problem arises when the authors conflate predictive modeling and fingerprinting. Again no one has suggested fingerprints are useful for personalized medicine – they simply provide a measure of the uniqueness and reliability of the individual connectome.

Line 71, 72 – The field already has an alternative perspective of the relationship between fingerprinting and prediction.

Line 287, 288 – No one has suggested that participant identification and behavioral prediction truly rest on the same functional connectome signatures.

Line 415, 416 – No one has argued that there is a relationship between the edges revealed in prediction and fingerprinting.

Line 472 – There are no reports linking connectome fingerprinting signatures directly to behavior that this reviewer is aware of.

Point-by-point response to editor and reviewer comments:

Reviewer #1 (Remarks to the Author):

(1.1) This a compelling article aiming to replicate the previous seminal work on fingerprinting (Finn et al. Nat Neuro 2015) and extend it to other tests (grip strength, language comprehension) that are not associated with fluid IQ used in the original report.

The authors used the unrelated HCP dataset (n=339) and four parcellation schemes (atlases: Shen/Brainnetome/HCP MMP 1.0/AAL), two prediction methods, and tested different feature selection thresholds. The authors were able to replicate the original findings, albeit with a lower significance which they accurately address in the limitations. They were further able to show that connectivity patterns to identify individuals (discriminatory signatures) and to predict behaviour (predictive signatures) diverge and therefore conclude that there is no linear one-to-one association between behaviour and fingerprints.

The manuscript follows good practice in terms of open science and offers their codes online.

We thank the reviewer very much for the positive evaluation of our article.

(1.2) A minor comment would be that the healthy volunteers should not be referred to as 'subjects'.

In response to this comment, we replaced all references to healthy volunteers as "subjects" with the term "participants".

Reviewer #2 (Remarks to the Author):

(2.1) This manuscript examines the relationship between the functional connectivity features that are unique to an individual (ala functional fingerprinting) and those features that predict behavior. The methods are generally robust, and the investigators have done a thorough job comparing across parcellation schemes, thresholds, and prediction algorithms, and performing cross-validation where possible.

We thank the reviewer for their extensive and constructive feedback, which allowed us to significantly improve the outline, framing and scope of our manuscript.

(2.2) The basic conclusion is that connectome fingerprints are not as functionally relevant as previously believed. But the "previously believed" part unfortunately represents a misreading of the literature.

To this reviewer's knowledge there is nothing in the literature that states that the connections that make an individual unique and identifiable (the functional fingerprinting part) are supposed to be the same connections linking functional organization to behavior (the prediction part).

We thank the reviewer for this helpful comment. While we agree that the literature does not explicitly state that unique and predictive connections necessarily are the same, we also believe that the original Finn article can be interpreted as suggesting that both are related, e.g., here:

"These results [fingerprinting and CPM relying on the same resting-state networks] reinforce the functional relevance of our identification analyses, in that the networks most discriminating of individuals [i.e. fingerprinting] are also the most relevant to individual differences in behavior [i.e. CPM]." (Finn et al., 2015)

We agree that the original Finn article does not suggest that individual features should be the same within those networks; however, we believe that the authors use the term 'functional relevance' to suggest some

relation between the two. Therefore, we now more clearly state that previous data suggests a relation between fingerprinting and behaviour and that this relation has not yet been thoroughly investigated (see line 57-62). We believe that references to functional relevance of fingerprinting features such as quoted above, warrant a detailed investigation into potential overlaps on different levels of network granularity and organization.

Consequently, we have tempered statements about previous claims in the literature and instead focused the manuscript on characterizing the precise relationship between fingerprinting and CPM features (line 21-23 and line 41-44)).

(2.3) Fingerprinting reveals that the connectivity matrices for individuals are both stable and unique to individuals (a finding that had been unexpected and not previously recognized). The Finn 2015 paper showed this. They also demonstrated that within this connectome there is information on the individual and connectome-based modeling supported this claim. These are 2 separate findings. Finn did show that the same networks fronto-parietal and default mode dominated in both fingerprinting and prediction but there was no claim that these were, or should, be the same features within those networks.

We agree with the reviewer's analysis of fingerprinting and connectome-based modeling being two separate findings. However, similar to above (2.2.) we believe that data (from Finn et al., 2015 and others) *suggest* a relation between fingerprinting and behaviour that has not yet been thoroughly investigated. Therefore, we now state this more clearly in the introduction (see line 54-62).

(2.4) I don't believe that a close reading of the literature reveals any suggestions that the connections that yield brain fingerprints are the same as the connections that provide information on behavior.

With respect to the literature on fingerprinting, we believe that the functional relevance of fingerprinting features remains actively debated. Finn et al., (2021) just published an opinion piece discussing the relation between fingerprinting and predictive connectomes, citing- as part of an ongoing discussion – our preprint. As an additional example, Mansour et al. (2021) state:

“The ultimate goal of neural fingerprinting is not solely to identify one individual from a group of others, but to also appropriately link to individual behavioral differences (Finn et al., 2017). Nevertheless, the extent to which this identifiable information is associated with behavior is not understood.”

It is in this context, that we believe our analysis of the distinctive signatures in fingerprinting is ultimately warranted. To us, the “appropriate link to individual behaviour” is an open question as mentioned by Mansour et al. (2021) and is still actively debated in the field (e.g. Talk by Finn at Brain Connectivity Workshop 2021, <https://bcw-2021.com>, talk to be uploaded soon) that we herewith investigate. We hope that our re-framing of the issue (line 21-22, 54-62) and line is clear and helpful for the reader.

(2.5) Fingerprints have never been proposed as a method for personalized medicine. Behavioral prediction supports the finding that connections vary across individuals in meaningful ways and may someday make individualized medicine possible. This says nothing about fingerprinting and should not be conflated with fingerprinting. The personalized medicine part comes in the sense that individuals can be characterized through predictive models. This work does not dispute that.

We thank the reviewer for the helpful comment. After some deliberation, we agree that our mention of personalized medicine was unwarranted and have removed it entirely (line 35-36, 41-44).

(2.6) The authors have the correct and relevant explanation for why fingerprinting and predictive modeling

are different as they explain in the discrimination/variability discussion in lines 428-437. These are different methods not designed to reveal the same features – and they don't.

We agree with the reviewer and thank them for the comment. As mentioned above, we believe our discussion (now in lines 444-452) is still warranted given the suggestive nature of the previous data towards a relation between the two features. As this relation is not a clear one, a discussion of the different features revealed by each method still seems appropriate to us.

(2.7) The thorough investigation here of the edges involved in the models and in identification is potentially interesting as is the edge-level variability work, and the node topography findings. But the current framing of the manuscript represents a misreading of the literature.

We again thank the reviewer very much for bringing these issues to our attention. In response to the reviewer's comments, we thoroughly reframed our manuscript with different changes in abstract, introduction and discussion. Changes now read:

Abstract:

“One method that harnesses such individual signatures is functional connectome fingerprinting, which can reliably identify individuals from large study populations. However, the precise relationship between functional signatures underlying fingerprinting and behavioural prediction remains unclear.” (p. 2, first paragraph)

As well as:

Introduction:

“The method can be used as a measure of the uniqueness and reliability of individual functional connectomes (Amico & Goñi, 2018; Horien, Shen, Scheinost, & Constable, 2019; Milham, Vogelstein, & Xu, 2021).

Interestingly, networks most discriminating between individuals were also found to be most relevant to individual differences in cognitive performance and behaviour, raising the possibility that connectome fingerprints may also be functionally relevant (Finn et al., 2015; Liu, Liao, Xia, & He, 2018, Mansour, Tian, Yeo, Croypley, & Zalesky, 2021). While these previous findings were suggestive of such a relationship, they were also restricted to visual inspection on a network level, with a robust statistical analysis still missing.” (p. 3, second paragraph).

As well as:

Discussion:

“These findings could be interpreted as supporting the functional relevance of fingerprinting signatures, that is, networks that best discriminate individuals from one another are also strongly involved in cognitive function (Finn et al., 2015, 2017). However, these findings were restricted to a specific level of analysis (group-level within-network connections), motivating further exploration of this relationship.” (p. 22, first paragraph)

We hope that this reframing makes clear that the motivation for our investigation of distinctive and predictive signatures is based on our assumption that functional relevance should include overlap or correlation of features on multiple levels of analysis. While the seminal Finn et al., 2015 paper served as an initial impetus, the main motivation of our work was a detailed investigation into the functional relevance of fingerprinting, which was still lacking in the literature.

(2.8) This reviewer disagrees with the assumptions behind the following statements:

Line 33 – No one has suggested that connectome fingerprints are the functionally relevant features.

See next comment, because the mentioned lines are consecutive. For discussion of ‘functional relevance’, please see our comments above, specifically 2.2.

(2.9) Line 34, 35 – The results in this manuscript support that individual behavioral prediction is supported by connectome based predictive modeling and thus no re-evaluation is necessary. The problem arises when the authors conflate predictive modeling and fingerprinting. Again no one has suggested fingerprints are useful for personalized medicine – they simply provide a measure of the uniqueness and reliability of the individual connectome.

We thank the reviewer for the comment and we have now sharpened our manuscript by removing all mentions of personalized medicine (line 35-36, 41-44) and reframing the motivation for our investigation. We do still suggest that an investigation of the functional significance of fingerprinting is warranted, as the functional relevance was underscored in the initial paper by Finn et al., 2015 as well as currently in the literature e.g., in Mansour et al., 2021 (see above, comments 2.2 and 2.4).

Line 32-35 now reads: These results demonstrate that participant identification and behavioural prediction involve highly distinct functional systems of the human connectome. The present study thus calls into question the functional significance of functional connectivity fingerprints.

Previous Line 32-35: These results demonstrate that participant identification and behavioural prediction involve highly distinct functional systems of the human connectome, suggesting that connectome fingerprints are not as functionally relevant as previously believed. The present study thus calls for a re-evaluation of the significance of functional connectivity fingerprints in personalized medicine.

(2.10) Line 71, 72 – The field already has an alternative perspective of the relationship between fingerprinting and prediction.

We agree that there are also alternative perspectives of the relationship between fingerprinting and prediction and therefore propose our perspective as an additional one, based on edge-level variability, which reviewer #2 also regarded as interesting (see comment 2.6).

Line 71 now reads: Together, our results suggest an additional perspective on the relation between fingerprinting and behavioural prediction that rests on edge-level variability.

Previous Line 71: Together, our results suggest an alternative perspective on the relation between fingerprinting and behavioural prediction that rests on edge-level variability.

(2.11) Line 287, 288 – No one has suggested that participant identification and behavioral prediction truly rest on the same functional connectome signatures.

We agree with the reviewer and reframed the according paragraph similar to our argument in comments 2.2 and 2.4. While authors are not explicit in stating that the ‘signatures are the same’, discussions of functional relevance of connectome fingerprints suggest some relation might be expected. Therefore, characterizing the precise relationship between functional connectome signatures and behavioural prediction warranted more detailed investigation, motivating our study. Furthermore, we would expect an additional relationship between the signatures (beyond the resemblance of within-network contributions), if the participant identification was directly related to behaviour (as we believe the excerpts in 2.2 and 2.4 suggest).

Line 287 now reads: The above findings notwithstanding, if participant identification analysis is to be functionally relevant, some degree of relationship between discriminatory and predictive features would be expected beyond the mere resemblance of within-network contributions, e.g., overlap at the level of single edges, between-network connections, or the large-scale spatial distribution of discriminatory and predictive nodes.

Previous Line 287: The above findings notwithstanding, if participant identification and behavioural prediction truly rest on the same functional connectome signatures, significant overlap between discriminatory and predictive features would be expected beyond the mere resemblance of within-network contributions, i.e., at the level of single edges, between-network connections, and the large-scale spatial distribution of discriminatory and predictive nodes.

(2.12) Line 415, 416 – No one has argued that there is a relationship between the edges revealed in prediction and fingerprinting.

We thank the reviewer for this comment. We have removed the relevant section and reframed the according statement. As it is framed now, we present the discovered divergence as a fact relevant to our investigation of the relation between connectome signatures.

Line 415, 416 now reads: Here, we found evidence of a strong divergence between functional signatures supporting prediction of behaviour and connectome fingerprints.

Previous Line 415, 416: Here, we found evidence to the contrary, as there was a strong divergence between functional signatures supporting prediction of behaviour and connectome fingerprints.

(2.13) Line 472 – There are no reports linking connectome fingerprinting signatures directly to behavior that this reviewer is aware of.

We thank the reviewer and we have removed the relevant statement.

Line 472 now reads: Here, we show in detail that participant identification and behavioural prediction from individual connectomes rely on highly distinct functional brain systems.

Previous Line 472: In contrast to initial reports linking connectome fingerprinting signatures directly to behaviour, we here show in detail that participant identification and behavioural prediction from individual connectomes rely on highly distinct functional brain systems.

Finn, E. S. & Rosenberg, M. D. Beyond fingerprinting: Choosing predictive connectomes over reliable connectomes. *NeuroImage* 118254 (2021).

Finn, E. S. *et al.* Can brain state be manipulated to emphasize individual differences in functional connectivity? *NeuroImage* **160**, 140–151 (2017).

Finn, E. S. *et al.* Functional connectome fingerprinting: identifying individuals using patterns of brain connectivity. *Nature Neuroscience* **18**, 1664–1671 (2015).

Mansour L, S., Tian, Y., Yeo, B. T. T., Cropley, V. & Zalesky, A. High-resolution connectomic fingerprints: Mapping neural identity and behavior. *NeuroImage* **229**, 117695 (2021).

Reviewers' comments:

Reviewer #1 (Remarks to the Author):

I thank the reviewers for addressing my comments and thoroughly enjoyed reading the rebuttal between the authors and the second reviewer. Very insightful discussion.

Reviewer #2 (Remarks to the Author):

This work compares the networks associated with functional fingerprinting with the networks that predict behavior and concludes that connectome fingerprints are not as functionally relevant as previously believed.

It's not clear where the authors got the notion that functional fingerprints are supposed to be functionally relevant or used in individualized medicine.

Admittedly, the term functional fingerprinting is becoming over-used, but, when it was first introduced by Finn et al 2015 it was used to demonstrate that the connectome is unique to individuals and stable across sessions – sufficiently so that individuals could be identified. A second component of that work was to demonstrate that the connectome contains information related to fluid intelligence and subsequent work by that group and others developed means to relate the connectome to a range of behaviors and clinical variables.

These were separate concepts that the authors have now conflated.

It is true that Finn et al showed the fronto-parietal and default mode networks were most involved in both fingerprinting and in predictive models relating the connectome to fluid intelligence, there was no claim that these are the same edges, and as the authors correctly point out this was a simple visual observation.

In line 287 the authors write "The above findings notwithstanding, if participant identification and behavioral prediction truly rest on the same functional connectome signatures, significant overlap between discriminatory and predictive features would be expected...". Again, this concept that they should have the same connectome signatures represents a misreading of the literature. The authors modeled 30 behaviors, and one might imagine 100s of behaviors could be modeled, and if applied clinically 100's of symptoms or clinical scores could be modeled. Each of these models should involve different (but possibly overlapping) networks. To build a predictive model it is necessary to find edges that reflect the same behavior in varying degree across subjects – by definition, these cannot be totally unique edges – they have to follow the same pattern as that found in other individuals. It is not expected that the unique edges that define a subject are the same as the ones that define their behavior. The authors point this out in lines 431-440.

The overall methodology of this work – replicating the fingerprinting approach and connectome modeling of Finn et al is solid.

While the authors tested various thresholds in the predictive modeling frame-work the primary analysis did not include the effect of threshold in the assessment of overlap. In that test they applied arbitrary thresholds (99th percentile edge DP, and 80% of CV folds for the predictive edges). Similarly, to assess the overlap of fingerprinting and high variability edges the DP matrix was overlaid with a connectivity standard deviation matrix across subjects thresholded at 99th percentile. The results at lower thresholds could also be part of the primary analysis.

Reviewer #3 (Remarks to the Author):

Disclosure: I'm a new reviewer and I was given the initial reviews for this manuscript, but to prevent any bias I read the manuscript and performed this review prior to reading the first round of reviews by the other reviewers. My background is statistics with a specialty in the analysis of fMRI data.

The general idea of this work is to try and quantify how much an edge in a network contributes to fingerprinting and then whether that same edge has predictive ability in the case of predicting behavior. The case the authors are trying to make is that the connections driving fingerprinting may not be the same as those driving prediction. The biggest issue with the analysis is that it isn't clear why the metrics used to measure strength in prediction for fingerprinting evaluation would be comparable to the metrics used in the prediction of behavior and whether this is a common mistake in the field that needs correcting. Has that claim been made by others? If so, this work gains a bit of traction and that needs to be described more with references. Otherwise I'm not sure it is clear why the overlap would be necessary. Although this doesn't perfectly parallel what you're trying to show here, but there are many examples in fMRI-based voxelwise prediction where the SVR weights (for voxels as features) do not overlap with the univariate analyses. An example can be found in Chevalier et al (2021) as well as explanations for why one wouldn't necessarily expect those maps to match. There are also parallels between ICC between brain measures in two sessions as an indication of a limitation of that brain measure to correlate with behavior (Elliott et al, 2020), but in that case the link between ICC and the correlation is theoretical. Here it isn't clear to me that I'd expect a link between differential power and support vector regression weights for a variety of reasons. DP is calculated in a marginal type analysis (focusing on one edge at a time), whereas the SVR is a conditional analysis, such that a given feature must add something beyond what the other features are doing. Importantly, feature weights are a reflection of the data and model and the story can change with a different type of classifier or regularization strategy, so these results really only reflect prediction with SVR and it isn't necessarily the case that the same features would drive other prediction models (if the feature weights are interpretable).

The DP measure is not described in this work. I did look it up in Finn and I see it is attempting to isolate which edge is contributing to the stronger correlation across all edges between two sessions of a single subject. There is no motivation why this would indicate high predictive performance or whether it has previously been interpreted in this way.

It isn't clear how the edges were "selected" for the SVR. Were you thresholding or selecting based on whether or not it was zero. How correlated are the features?

Admittedly the specifics of connectivity fingerprints are new to me, but it seems that the fingerprint is a connectivity pattern within-subject and it has been given the name "fingerprint" because it appears to be somewhat unique to that subject. On the other hand the DP metric was an attempt to quantify where the differences were occurring in the "fingerprints". I feel like these two ideas are often used interchangeably in the manuscript, which is confusing. For example, line 325 on p 8 explains that predictions based on discriminatory edges weren't good at predicting, reflecting that it is a bad idea to use DP for feature selection. Yet it then says, "further corroborating that individual fingerprints are not related to behavior on a single edge level". That's a confusing statement. If the full set of edges predicted well, then it seems the "fingerprint" is good at predicting. I'm unsure what is meant by single edge level.

Chevalier et al (2021). Decoding with confidence: Statistical control in decoder maps. *NeuroImage*
Elliott et al (2020). What is the test-retest reliability of common task-functional MRI measures? *Psych Science*.

Point-by-Point Response

Reviewer #1 (Remarks to the Author):

I thank the reviewers for addressing my comments and thoroughly enjoyed reading the rebuttal between the authors and the second reviewer. Very insightful discussion.

We again thank R#1 for their comment.

Reviewer #2 (Remarks to the Author):

This work compares the networks associated with functional fingerprinting with the networks that predict behavior and concludes that connectome fingerprints are not as functionally relevant as previously believed.

It's not clear where the authors got the notion that functional fingerprints are supposed to be functionally relevant or used in individualized medicine.

We thank the reviewer for this comment. We do, however, believe that we addressed this point thoroughly in our previous point-by-point response (previous point-by-point 2.2 and 2.4), where we stated the following:

(2.2) The basic conclusion is that connectome fingerprints are not as functionally relevant as previously believed. But the “previously believed” part unfortunately represents a misreading of the literature.

To this reviewer's knowledge there is nothing in the literature that states that the connections that make an individual unique and identifiable (the functional fingerprinting part) are supposed to be the same connections linking functional organization to behavior (the prediction part).

We thank the reviewer for this helpful comment. While we agree that the literature does not explicitly state that unique and predictive connections necessarily are the same, we also believe that the original Finn article can be interpreted as suggesting that both are related, e.g., here:

“These results [fingerprinting and CPM relying on the same resting-state networks] reinforce the functional relevance of our identification analyses, in that the networks most discriminating of individuals [i.e. fingerprinting] are also the most relevant to individual differences in behavior [i.e. CPM].” (Finn et al., 2015)

We agree that the original Finn article does not suggest that individual features should be the same within those networks; however, we believe that the authors use the term ‘functional relevance’ to suggest some relation between the two. Therefore, we now more clearly state that previous data suggests a relation between fingerprinting and behaviour and that this relation has not yet been thoroughly investigated (see line 57-62). We believe that references to functional relevance of fingerprinting features such as quoted above, warrant a detailed investigation into potential overlaps on different levels of network granularity and organization. Consequently, we have tempered statements about previous claims in the literature and instead focused the manuscript on characterizing the precise relationship between fingerprinting and CPM features (line 21-23 and line 41-44).

(2.4) I don't believe that a close reading of the literature reveals any suggestions that the connections that yield brain fingerprints are the same as the connections that provide information on behavior.

With respect to the literature on fingerprinting, we believe that the functional relevance of fingerprinting features remains actively debated. Finn et al., (2021) just published an opinion piece discussing the relation between fingerprinting and predictive connectomes, citing- as part of an ongoing discussion – our preprint. As an additional example, Mansour et al. (2021) state:

“The ultimate goal of neural fingerprinting is not solely to identify one individual from a group of others, but to also appropriately link to individual behavioral differences (Finn et al., 2017). Nevertheless, the extent to which this identifiable information is associated with behavior is not understood.”

It is in this context, that we believe our analysis of the distinctive signatures in fingerprinting is ultimately warranted. To us, the “appropriate link to individual behaviour” is an open question as mentioned by Mansour et al. (2021) and is still actively debated in the field (e.g. Talk by Finn at Brain Connectivity Workshop 2021, <https://bcw-2021.com>, talk to be uploaded soon) that we herewith investigate. We hope that our re-framing of the issue (line 21-22, 54-62) and line is clear and helpful for the reader.

Admittedly, the term functional fingerprinting is becoming over-used, but, when it was first introduced by Finn et al 2015 it was used to demonstrate that the connectome is unique to individuals and stable across sessions – sufficiently so that individuals could be identified. A second component of that work was to demonstrate that the connectome contains information related to fluid intelligence and subsequent work by that group and others developed means to relate the connectome to a range of behaviors and clinical variables.

These were separate concepts that the authors have now conflated.

We kindly disagree with this assessment and we had tried to point out more clearly where in the literature we believe these two concept to be conflated (see also 2.2 and 2.4 copied above). Indeed, we believe that already in the initial Finn paper (Finn et al., 2015) the mentioned conflation is present, justifying our empirical investigation of the relation of fingerprinting and behavioural prediction. We have addressed these concerns already in the previous revision of the manuscript.

We are, therefore, not sure how to address this concern any further, since the reviewer seems not haven taken into account our previous replies to this issue and the according changes in the manuscript.

It is true that Finn et al showed the fronto-parietal and default mode networks were most involved in both fingerprinting and in predictive models relating the connectome to fluid intelligence, there was no claim that these are the same edges, and as the authors correctly point out this was a simple visual observation.

While it is true that Finn et al. did not mention edge-to-edge correspondence an instead based their conclusion on a simple visual observation, these findings were used to argue for the *“functional relevance of our identification analyses, in that the networks most discriminating of individuals are also the most relevant to individual differences in behaviour”* (Finn et al., 2015). We then follow up (empirically) on this observation, believing that *functional relevance* should require more than simple visual observation and discover that – apart from the visual overlap – there seems to be little in common between highly distinctive edges and highly predictive edges.

In line 287 the authors write “The above findings notwithstanding, if participant identification and behavioral prediction truly rest on the same functional connectome signatures, significant overlap between discriminatory and predictive features would be expected...”. Again, this concept that they should have the same connectome signatures represents a misreading of the literature. The authors modeled 30 behaviors, and one might imagine 100s of behaviors could be modeled, and if applied clinically 100’s of symptoms or clinical scores could be modeled. Each of these models should involve different (but possibly overlapping) networks. To build a predictive model it is necessary to find edges that reflect the same behavior in varying degree across subjects – by definition, these cannot be totally unique edges – they have to follow the same pattern as that found in other individuals. It is not expected that the unique edges that define a subject are the same as the ones that define their behavior. The authors point this out in lines 431-440.

Unfortunately, the line quoted above (line 287) had been removed in the revised manuscript. The fact that R#2 cites it here, suggests to us that they did not read the revised manuscript, where we had removed and changed the line in accordance with R#2s initial comments.

The overall methodology of this work – replicating the fingerprinting approach and connectome modeling of Finn et al is solid.

While the authors tested various thresholds in the predictive modeling framework the primary analysis did not include the effect of threshold in the assessment of overlap. In that test they applied arbitrary thresholds (99th percentile edge DP, and 80% of CV folds for the predictive edges). Similarly, to assess the overlap of fingerprinting and high variability edges the DP matrix was overlaid with a connectivity standard deviation matrix across subjects thresholded at 99th percentile. The results at lower thresholds could also be part of the primary analysis.

We believe that we did address this concern in the manuscript (however, we are not 100 % sure to correctly understand the issue here): in the supplementary material, we test whether our overlap results hold with different thresholds (.001, .005, .01, .05) for both DP and predictive edges (see Supplementary Figure 2 and 3). We tested different thresholds for the variability overlap as well (99th, 98th and 95th percentile), which can be seen in Figure 4, left. In order to more clearly point the reader to this information, we have now added direct pointers to relevant supplementary material (e.g., line 217 - 219)

Reviewer #3 (Remarks to the Author):

1.1:

Disclosure: I'm a new reviewer and I was given the initial reviews for this manuscript, but to prevent any bias I read the manuscript and performed this review prior to reading the first round of reviews by the other reviewers. My background is statistics with a specialty in the analysis of fMRI data.

We want to thank R#3 for their thoughtful review and analysis, which allowed us to substantially improve our manuscript. We agree with their main remarks, namely questions about the motivation of our study,

their methodological perspective on the relation of prediction and identification, an issue with conceptual clarity and information missing from our prediction pipeline using SVR and CPM.

1.2:

The general idea of this work is to try and quantify how much an edge in a network contributes to fingerprinting and then whether that same edge has predictive ability in the case of predicting behavior. The case the authors are trying to make is that the connections driving fingerprinting may not be the same as those driving prediction. The biggest issue with the analysis is that it isn't clear why the metrics used to measure strength in prediction for fingerprinting evaluation would be comparable to the metrics used in the prediction of behavior and whether this is a common mistake in the field that needs correcting. Has that claim been made by others? If so, this work gains a bit of traction and that needs to be described more with references.

We agree with R#3s comment and point out that we removed significant parts of our motivation in response to R#2s remarks, which have likely clouded our initial argument. It should be noted that even though we didn't agree with the criticism of our motivation by R#2 that have led to the changes in our 2nd resubmission, we nevertheless followed their advice to accommodate their concerns (see previous point-by-point response). We therefore strongly agree with R#3 that our motivation needs to be clarified (even though going against R#2s original criticism) and have now corrected this by restating our argument in the introduction.

We now clearly state that the literature has treated connectome fingerprints as functionally relevant. While the literature does not explicitly state that unique and predictive connections necessarily are the same, we believe that the original Finn article suggests they are related, e.g., here:

“These results [fingerprinting and CPM relying on the same resting-state networks] reinforce the functional relevance of our identification analyses, in that the networks most discriminating of individuals [i.e. fingerprinting] are also the most relevant to individual differences in behavior [i.e. CPM].” (Finn et al., 2015)

The original Finn article does not state that individual edges in these networks must be identical; however, the authors use the term ‘functional relevance’ to suggest some relation between behaviour (here investigated using prediction) and fingerprinting. This relation has also been stated by others (e.g., Mansour et al., 2021) and implicitly assumed by others still (Liu et al., 2018), but has so far not been thoroughly investigated. We believe that references to functional relevance of fingerprinting features such as quoted above, warrant a detailed investigation into potential overlaps on different levels of network granularity and organization.

We have now restated the relevant introductory paragraph (line 50 – 65) to read:

Connectome fingerprinting represents one such individualised and powerful approach to single-subject analysis (Finn et al., 2015, 2017; Kaufmann et al., 2017; Emerson et al., 2017; Waller et al., 2017; Schulz et al., 2020; Lin, Baete, Wang, & Boada, 2020;). In connectome fingerprinting, individual participants can be reliably identified within large datasets with accuracies exceeding 90%, based on the discriminatory power of individual functional connectomes. While the method can be used as a measure of the uniqueness and reliability of individual functional connectomes (Amico & Goñi, 2018; Horien, Shen, Scheinost, & Constable, 2019; Milham, Vogelstein, & Xu, 2021), its large appeal to the community is likely more practical: the possibility of a relation between connectome fingerprinting and behaviour, individual traits or clinical markers (Mansour, Tian, Yeo, Copley, & Zalesky, 2021). This assumption that distinctive features are functionally relevant (e.g., Liu, Liao, Xia, & He, 2018; Lin, Baete, Wang and Boada, 2020; Cai et al., 2021)

rests on the report of networks most discriminating between individuals also being those most relevant to individual differences in cognitive performance and behaviour (Finn et al., 2015). However, while these previous findings were suggestive of a relationship between distinctive and predictive features of the functional connectome, the analysis was restricted to visual inspection on a network level with a robust statistical analysis still missing.

1.3:

Otherwise I'm not sure it is clear why the overlap would be necessary. Although this doesn't perfectly parallel what you're trying to show here, but there are many examples in fMRI-based voxel-wise prediction where the SVR weights (for voxels as features) do not overlap with the univariate analyses. An example can be found in Chevalier et al (2021) as well as explanations for why one wouldn't necessarily expect those maps to match. There are also parallels between ICC between brain measures in two sessions as an indication of a limitation of that brain measure to correlate with behavior (Elliott et al, 2020), but in that case the link between ICC and the correlation is theoretical. Here it isn't clear to me that I'd expect a link between differential power and support vector regression weights for a variety of reasons. DP is calculated in a marginal type analysis (focusing on one edge at a time), whereas the SVR is a conditional analysis, such that a given feature must add something beyond what the other features are doing.

We agree with R#3 that the methodological differences between connectome fingerprinting and prediction (be it SVRs or CPM) don't clearly suggest why they should be related. In other words: We agree with R#3 in that there is no good a priori argument for a link between discriminatory edges and predictive edges. However, the literature is not clear on the relation (see 1.2 and our changed introduction), and we believe the possibility of a link to be one of the reasons fingerprinting has become so popular in recent years with the original manuscript gaining over 1000 citations within 6 years. Therefore, we empirically investigate it and in the discussion state why we think the link between discriminatory and predictive networks does not hold.

1.4.:

Importantly, feature weights are a reflection of the data and model and the story can change with a different type of classifier or regularization strategy, so these results really only reflect prediction with SVR and it isn't necessarily the case that the same features would drive other prediction models (if the feature weights are interpretable).

We agree with R#3 that the specific features selected / feature weights depend heavily on the data at hand and on the model, that is classifier or regularization strategy. In fact, the reason we used SVR was as an additional – confirmatory – analysis. That is, we used SVR in order to show that our results hold in different frameworks with different prediction models (resulting in varying feature weights between prediction algorithms). The focus in our main analysis was on CPM (Connectome-based predictive modeling), i.e., the prediction method that was previously employed by Finn, from which the purported relation between fingerprinting and prediction was argued (Finn et al., 2015). In our case, both CPM in the main analysis as well as SVR as confirmation, showed the same point: that predictive and highly discriminatory edges are not related and both prediction strategies resulted in the dichotomy based on edge variability (see Fig. 4). So, while we agree that our analysis does not necessarily generalize across all possible feature selection or classification methods, we propose that the use of CPM is – in our case – the relevant one, further corroborated by our additional SVR-based analysis.

We clarified this by changing parts of our method section (lines 197 - 199), which now reads:

Lastly, we used an additional prediction method (support vector regression) to evaluate, whether the observed overlap of discriminatory and predictive edges held using different classifier types, independently of CPM.

1.5:

The DP measure is not described in this work. I did look it up in Finn and I see it is attempting to isolate which edge is contributing to the stronger correlation across all edges between two sessions of a single subject. There is no motivation why this would indicate high predictive performance or whether it has previously been interpreted in this way.

We thank the reviewer for their comment and believe that this topic is also treated in 1.2.

1.6:

It isn't clear how the edges were "selected" for the SVR. Were you thresholding or selecting based on whether or not it was zero. How correlated are the features?

We thank R#3 for pointing out that the description of our SVR pipeline was insufficiently clear. In response, we added the following description how the features were selected (line 201 - 205) to our methods section:

SVR parameters were set at default values with no hyperparameter optimization (linear kernel, $C = 0.75$, Regularization = Lasso, $\Lambda = 0.0035$). We extracted the highly predictive edges using the sorted SVR weights, which were thresholded and binarized for the subsequent overlap analysis. Using the same method applied to the PP edges, all edges selected in at least 80% of CV folds during prediction were used, resulting in binarized matrix (described below).

1.7:

Admittedly the specifics of connectivity fingerprints are new to me, but it seems that the fingerprint is a connectivity pattern within-subject and it has been given the name "fingerprint" because it appears to be somewhat unique to that subject. On the other hand the DP metric was an attempt to quantify where the differences were occurring in the "fingerprints". I feel like these two ideas are often used interchangeably in the manuscript, which is confusing. For example, line 325 on p 8 explains that predictions based on discriminatory edges weren't good at predicting, reflecting that it is a bad idea to use DP for feature selection. Yet it then says, "further corroborating that individual fingerprints are not related to behavior on a single edge level". That's a confusing statement. If the full set of edges predicted well, then it seems the "fingerprint" is good at predicting. I'm unsure what is meant by single edge level.

We thank R#3 for their substantial and very thoughtful remark, which allowed us to significantly improve our manuscript. Thinking carefully about it, we realized that there was a conceptual ambiguity in our manuscript. We previously treated "individual fingerprints" to mean "highly distinctive parts of the functional connectome". While we still believe this understanding to be reasonable (fingerprints being the discriminatory individual signatures in the functional connectome that allow for successful fingerprinting, instead of fingerprints just being another word for "functional connectome"), we see how this may be confusing to the reader.

Therefore, instead of speaking of "individual fingerprints" as in the example above (now line 331), we changed this into "discriminatory edges" or "edges with high discriminatory potential" where appropriate.

Accordingly, we made the relevant changes also lines 23-25, 28, 67, 214-215, in table 2, table 3, lines 345-348, 387-388, figure 4, figure 5, line 437, 441-442 and line 530 and hope that these adjustments clear up the confusion.

Reviewers' comments:

Reviewer #1 (Remarks to the Author):

The authors did incorporate my suggestions and I have no further comments to the authors.

Reviewer #3 (Remarks to the Author):

I would like to thank the authors for their response.

Unfortunately I'm still a bit unclear as to whether, mathematically, DP and PP would be expected to be comparable. DP sort of tries to tease out what edges contribute to a correlation between two sets of edge weights, while PP focuses on a single edge's correlation between edge weight and behavior between-subject. I have no intuition as to why these two metrics would have large edge overlap. Although I wish more was done to motivate why (theoretically and mathematically) one would even expect the DP and PP measures to yield overlapping edges, I have no new comments.

Point-by-Point Response

Reviewer #3 (Remarks to the authors):

I would like to thank the authors for their response.

We also thank reviewer #3 for their response and hope that the inclusion of a more extensive justification for our investigation in the introduction as well as the adoption of some of their methodological criticism in the discussion helps to elucidate why an empirical investigation of the relation of discriminatory and predictive features is warranted.

Unfortunately, I'm still a bit unclear as to whether, mathematically, DP and PP would be expected to be comparable. DP sort of tries to tease out what edges contribute to a correlation between two sets of edge weights, while PP focuses on a single edge's correlation between edge weight and behavior between-subject. I have no intuition as to why these two metrics would have large edge overlap. Although I wish more was done to motivate why (theoretically and mathematically) one would even expect the DP and PP measures to yield overlapping edges, I have no new comments.

We thank the reviewer for their valuable comment. In response to this point, we have thoroughly restructured our introduction to explain our motivation more adequately as requested and added more extensive theoretical background.

The relevant section of the introduction now reads:

Since its conception, connectome fingerprinting has raised the intriguing question of whether distinctive individual connectivity signatures are also functionally relevant to variation in behaviour (Finn et al., 2015). Subsequent literature on connectome fingerprinting is ripe with parallel investigations of individual identifiability and individual prediction of behaviour, for instance using static functional connectivity (Amico & Goñi, 2018; Byrge & Kennedy, 2020; Mansour, Tian, Yeo, Cropley, & Zalesky, 2021), dynamical functional connectivity (Liu, Liao, Xia, & He, 2018), structural connectivity (Lin, Baete, Wang, & Boada, 2020) and structural features like cortical thickness or myelin (Mansour et al., 2021). Indeed, those resting-state networks that show highest inter-individual variability – such as the frontoparietal, default mode, and dorsal attention network (Mueller et al., 2013; Laumann et al., 2015; Gordon, Laumann, Adeyemo, & Petersen, 2017) – have also been shown to contain highly discriminatory features in fingerprinting (Finn et al., 2015; Miranda-Dominguez et al., 2014, 2017; Menon & Krishnamurthy, 2019; Demeter et al., 2020) and are at the same time commonly associated with behaviourally and clinically relevant variability (Corbetta & Shulman, 2002; Cole et al., 2013; Marek & Dosenbach, 2018). In consequence, discriminatory fingerprinting signatures and inter-individual variability in behaviour have consistently been interpreted or assumed to be related (e.g., Finn et al., 2015; Xu et al., 2016; Liu et al., 2018; Jalbrzikowski et al., 2020; Cai, Chen, Liu, Zhu, & Yu, 2020; Lin et al., 2020; Mansour et al., 2021).

However, such interpretations about the functional relevance of connectome fingerprinting commonly rely on visual inspection of network-level organisation (i.e., the relative distributions of predictive or discriminatory connections in different resting-state networks), whereas both connectome

fingerprinting as well as behavioural prediction ultimately rest on individual edges. As such, claims about a link between behaviour and fingerprinting draw upon aggregated data instead of the underlying region-to-region connections, paralleled by lacking analytical explanation of why such a relationship might exist. Therefore, while previous findings have revealed suggestive network-level patterns pointing to a potential overlap between discriminatory and predictive features of the functional connectome, a detailed, multi-layered statistical analysis is necessary to investigate if such a relationship can truly be verified empirically.

Instead of:

This assumption that distinctive features are functionally relevant (e.g., Liu, Liao, Xia, & He, 2018; Lin, Baete, Wang and Boada, 2020; Cai et al., 2021) rests on the report of networks most discriminating between individuals also being those most relevant to individual differences in cognitive performance and behaviour (Finn et al., 2015). However, while these previous findings were suggestive of a relationship between distinctive and predictive features of the functional connectome, the analysis was restricted to visual inspection on a network level with a robust statistical analysis still missing.

Regarding a mathematical motivation, we agree that the literature lacks a clear mathematical rationale, which we now more clearly point out in the introduction. It is important to add that the assumptions in the literature are based on empirical evidence only which led us to address the previous claims empirically. At the same time, we agree with reviewer #3 that any mathematical explanation or rationale of such a relationship would be highly dependent on the specific method and model underlying the calculation of DP and PP.

Moreover, the valid point about conditional / marginal type analysis raised by reviewer #3 is pertinent to our analysis using SVR while we also investigate CPM. In CPM, just as in the calculation of DP, the predictive edges are calculated one edge at a time, independently of each other. As such, we would argue that the difference between marginal type and conditional analysis does not explain the lack of overlap in this case. To address this, we added to the discussion:

In the framework of SVR, the feature selection is interdependent, i.e., the weight of a specific feature depends on the additional amount of information supplied beyond other features. Nevertheless, similarly to CPM, the edge variation also must be linearly related to variability in behaviour.

However, we agree that - generally – the method and model of prediction could significantly influence the specific edges selected during prediction, although the observed variability-based dichotomy of DP and PP edges may represent a more general phenomenon.